# Income inequality, voters' support for public spending and the size of the welfare state. A simple political model

**Ángel Solano-García** [ID] *

Universidad de Granada, Granada, Spain

* asolano@ugr.es

## Abstract

This paper studies the effect of income inequality and voters' support for public spending on the choice of size of the welfare state. Based on new empirical findings showing that preferences for taxation depend on the nature of the policies financed with tax revenues (Barnes 2015, Ballard-Rosa 2016, Roosma 2016, and Berens 2019) I build a Downsian two-party political competition framework in which voters differ in both income (rich or poor) and ideology (liberal or conservative). Government provides two types of public services: one that increase the size of the welfare state and other that does not. Liberal (conservative) voters only care about the public service that increase (do not increase) the size of the welfare state. I find that the decisive voter and the size of the welfare state depends on both the level of income inequality and voters' support for public spending. In particular, and different from the traditional models on redistributive politics (Romer 1975, Roberts 1977, and Meltzer 1981), I obtain that an increase in pre tax income inequality may reduce the size of the welfare state chosen by majority voting.

## 1 Introduction

Based on purely *homo economicus* preferences, traditional theories on redistributive politics claim that low-income voters vote for parties in favor of more redistributive policies while high-income voters vote for parties advocating low redistributive policies. However, voters might be affected by other non-redistributive policies and, as a consequence, the relationship between voters' income and their voting choice is not so straightforward (see [1] and citations therein). Nowadays, it is a fact that Rightist parties are winning seats in many parliaments of developed countries thanks to the support of poor, low-educated voters (see [2], for the changes in voting patterns over the last 75 years). To attract this type of vote, these parties campaign against immigration and in favor of greater expenditure on border enforcement and homeland security, painting immigrants as job stealers and criminals [3, 4]. Other policies that aim to defend nationalistic sentiments and traditional values place blame on various collectives, such as separatist movements, LGBT or feminist groups, are also commonly used for the same goal [5, 6]. Nevertheless, most of these conservatives parties share an opposite view, that

**Funding:** Angel Solano Garcia thanks financial support from FEDER/Junta de Andalucía-Consejería de Transformación Económica,

Industria, Conocimiento y Universidades/Grant B-SEJ-10-UGR20. https://www.juntadeandalucia.es/organismos/transformacioneconomicaindustriaconocimientoyuniversidades.html. The funders had no role in study design, data collection and analysis, decision to publish, or preparation of the manuscript.

**Competing interests:** The authors have declared that no competing interests exist.

is, their dislike of social welfare protection, which goes against the economic interests of the poor. To combat this policy, they often spread pessimistic opinions about the role of the state in redistributing income and claim that much tax revenue is wasted. All of this may reduce the support for government intervention among poor conservatives and convince them not to vote for a more redistributive option.

In this paper I study the effect of income inequality and voters' support for public spending on the choice of size of the welfare state. To do this, and based on new empirical findings showing that preferences for taxation depend on the kinds of policies that are financed with tax revenues [7–10], I assume that voters not only care about the size of public expenditure but also about the goals of the policies financed with it. In particular, I assume that there are two types of policy goals, one more liberal, such as the fight against poverty, and the other more conservative, such as homeland security. I use the simple Downsian two-party political competition framework whereby there are four types of voters, which depends, first, on whether they are rich or poor, according to their personal income level, and second, on whether they are liberal or conservative, according to their policy preferences. In general, I find that an increase in income inequality and an increase in conservatives' support for public spending makes the conservative poor voter more decisive in elections. Also I obtain that an increase in income inequality might not decrease the size of the welfare state as traditional models on redistributive politics predict [1, 11, 12].

Most democracies have a structure of government spending, consolidated over their history, that reflects voters' preferences and their changes over time. This structure does not alter dramatically over time and can be very different across countries. For instance, according to World Bank Statistics, the US devoted 3.4% of their GDP to military expenditure, while in Sweden it was only 1.1%, and the changes in these shares has been small over time in both countries (See all data at https://data.worldbank.org/indicator/MS.MIL.XPND.GD.ZS). My model presents this feature through an exogenous majority of liberal or conservative voters who determine the structure of public expenditure. However the size of such public expenditure remains endogenous and it is the output of political competition.

This paper contributes to giving an answer to the question of why the size of the welfare state is not increasing in income inequality in democracies, as claimed in [1, 11, 12], and provides a theoretical explanation of the "existence of a gradual process of disconnection between the effects of income and education on the vote" [13]. I find that the size of the welfare state depends crucially not only on the level of income inequality but also on the extent to which conservatives differ from liberals in preferences for public spending. In particular, the model converges toward traditional models on redistributive politics only if the support for public spending is alike between conservatives and liberals. However if conservatives' support for public spending is low in comparison with liberals, high income inequality is compatible with a moderate tax rate in a democracy with a majority of liberal voters. In this case political constituencies are based on ideology rather than income level (different from the traditional models) and the rich liberal voter becomes the decisive voter. This paper also complements the existing literature in political economics using models of multidimensional politics [14–16].

The rest of the paper is organized as follows. In Section 2, I describe the theoretical model and provide the political equilibrium. In Section 3, I offer some concluding remarks.

## 2 The model

Consider an economy populated by voters of mass equal to one. Voters care about private consumption ($c$) and about one of these two types of public services: one with more liberal purposes that increases the size of the welfare state ($G_L$) and another with more conservative

purposes ($G_C$). An example of the former would be a social welfare program to fight poverty, while border enforcement would be an example of the latter.

I assume that there are two social identities (or political cleavages) in regard to electoral behavior: income level and ideology. The former is supported by the standard literature on redistributive politics [1, 11, 12]. The latter is not so popular and needs justification. Although in the US there is a clear party identification by voters, in Europe this is not so obvious. According to the International Social Survey Programme (ISSP), in 2003 the most frequent groups Western European voters identified with were class and occupation (37 percent), followed by nation (24 percent), gender (18 percent), age (13 percent), and religion (8 percent). The model can be reformulated without altering the results and their implications by using class and occupation instead of income, and the level of nationalism instead of the level of conservatism.

To make things simple, I assume that there are four types of voters according to their income level and their preferences for public services: Rich-Conservative, Rich-Liberal, Poor-Conservative and Poor-Liberal. Their relative population sizes are represented by $n_{ij}$, where $i = R, P$ indicates if they are Rich or Poor, and $j = L, C$ stands for their Liberal or Conservative ideology, such that $n_{RC} + n_{RL} + n_{PC} + n_{PL} = 1$. It is assumed that no single group holds a majority of the population, that is $n_{ij} \in (0, 1/2)$. I assume that voters' utility function is as follows:

$$u_{ij}(c_{ij}, G_L, G_C) = c_{ij} + \Delta_j G_L^{\frac{1}{2}} + (1 - \Delta_j)\beta G_R^{\frac{1}{2}},$$

where $i = \{R, P\}$ and $j = L, C$ and

$$\Delta_j = \begin{cases} 1 & \text{if} \quad j = L \\ 0 & \text{if} \quad j = C \end{cases}$$

Liberals and conservatives only care about the public good that targets their respective preferences. Since I consider that conservatives are more reluctant to increase public spending than liberals, I assume that $0 < \beta < 1$. Hence, $\beta$ stands for the relative intensity of conservatives' support for public spending. Regarding this assumption, in the 2019 survey by Pew Research Center in the US, a large majority of Republicans (74%) responded that they preferred a smaller government with fewer services. Moreover, among Republicans, more conservative voters are more likely than those more moderate or liberal to express this preference (82% to 57%). In contrast, a majority of Democrats are more likely to say they prefer a bigger government with more services (67%).

Rich and poor voters are endowed with an income level $y_R > 0$ and $y_P > 0$, respectively, such that $y_R > y_P$. I define income inequality using the ratio $\frac{y_P}{y_R}$. The larger the ratio, the lower the level of income inequality. According to the data on the income distribution of the most developed countries, I consider that there is a majority of poor voters, that is $n_{PC} + n_{PL} > 1/2$. Total income in the economy is thus $Y = (n_{RC} + n_{RL})y_R + (n_{PC} + n_{PL})y_P$.

Government is formed as a result of a majority voting process. Government choices are a proportional income tax rate, $\tau \in (0, 1)$, and the size of each type of public good, $(G_L, G_R)$. In particular let $G > 0$ be the total public expenditure in both public goods and let $\alpha \in [\underline{\alpha}, \bar{\alpha}]$ be the share of total public expenditure devoted to finance public good with liberal purposes. For the sake of simplicity, I assume that $\underline{\alpha} \in (0, 1/2), \bar{\alpha} \in (1/2, 1)$, and $\underline{\alpha} + \bar{\alpha} = 1$. This means that if a big share of tax revenue is financing the public good with a liberal goal then a small share of tax revenue is financing the public good with conservative purposes and *vice versa*.

Assuming that the government budget constraint is balanced, it follows that:

$$G_L + G_R = \tau Y \Leftrightarrow \alpha G + (1 - \alpha)G = \tau Y$$

Therefore, due to the latter constraint, the government policy instruments become two instead of three. I choose $(\tau, \alpha)$ as the government policy instruments.

I assume that government is formed after an election in which two purely opportunistic parties compete. The timing of the political competition process is as follows: first, voters know their type. Second, voting takes place in two stages. In the first stage, each party proposes a level of income tax and then voters vote for the one that gives them a higher utility. Finally, in the second stage, each party proposes the shares of the tax revenue allocated to finance the different public goods and voters again vote for the one that gives them a higher utility (results remain the same in the case of interchanging voting stages). I adopt this two-stage voting process to avoid the problem of multidimensionality when aggregating preferences (see [17], for a similar formulation).

## 2.1 Voters' optimal policies and political equilibria

Before calculating the political equilibria I characterize voters' preferences on policies. To do that I obtain the optimal policy for both liberal and conservative voters.

Let us start with liberal voters. The First Order Condition (FOC) of the utility maximization problem is:

$$\frac{\partial u_{iL}}{\partial \tau} = -y_i + \frac{1}{2}Y\alpha G^{-\frac{1}{2}} = 0$$

$$\frac{\partial u_{iL}}{\partial \alpha} = G^{\frac{1}{2}} > 0$$

Thus the optimal policies for liberal voters are:

$$\tau_{iL}^*(\alpha) = \frac{\alpha Y}{4y_i^2}, \quad \text{and} \quad \alpha_L^* = \bar{\alpha} \tag{2.1}$$

Similarly, the FOC of the utility maximization problem for conservative voters is:

$$\frac{\partial u_{iC}}{\partial \tau} = -y_i + \frac{1}{2}Y(1-\alpha)\beta G^{-\frac{1}{2}} = 0$$

$$\frac{\partial u_{iC}}{\partial \alpha} = -\beta G^{\frac{1}{2}} < 0$$

Thus, the optimal policies for conservative voters are:

$$\tau_{iC}^*(\alpha) = \frac{\beta(1-\alpha)Y}{4y_i^2}, \quad \text{and} \quad \alpha_C^* = \underline{\alpha} \tag{2.2}$$

Both liberals' and conservatives' optimal tax rates (and therefore public expenditure) are increasing in the share of tax revenues financing their favorite public good and decreasing in their personal income. Also note that, among individuals in the same income group, conservative voters always prefer an equal or lower tax rate than liberals and, in particular,

$$\tau_{iC}^*(\underline{\alpha}) = \beta\tau_{iL}^*(\bar{\alpha}) \tag{2.3}$$

Another interesting remark is that the model collapses into a traditional model in which ideology is not included if $\beta = 1$. In this case, there are only two optimal tax rates instead of four: the optimal tax rate for the poor and the optimal tax rate for the rich. In this scenario the poor is always the decisive voter so their favorite policy will be implemented in equilibrium. A rise in income inequality would lead to an increase in the tax rate implemented as well as in the size of the welfare state.

Once I have analyzed voters' policy preferences, I proceed to calculating the political equilibrium. Given that I have a two-stage game, I use the subgame perfect Nash equilibrium as the solution concept. Solving backwards, the policy selected in equilibrium depends on the number of liberal versus conservative voters in Stage 2. I present the equilibria for the scenario where there is a liberal majority. I do this because I believe it is the most realistic scenario. According to the European Social Survey there exist a strong public support for government intervention in the protection of the welfare state in all European countries [18]. The International Social Survey Programme also obtains the same result in a large number of European and non European countries [19]. The results for a conservative majority are presented in the S1 Appendix. However, the intuition of results in this scenario will be commented in the main text.

If the number of liberals is higher than the number of conservatives $n_{RL} + n_{PL} \geq 1/2$, both parties choose the optimal share of tax revenues for liberal voters in equilibrium, which is $\bar{\alpha}$. This is the scenario in which public resources are mostly used to accomplish liberal goals such as social protection. The calculus of the political equilibrium over the platforms on the tax rate is less straightforward. The following lemma characterizes the two possible orders of voters' optimal tax rate depending on the level of income inequality and the differences in voters' support for government intervention.

**Lemma 1** *There are two possible orders of voters' optimal tax rate, depending on the level of income inequality,* $\frac{y_P}{y_R}$, *and the differences in voters' support for government intervention,* $\beta$.

- *Case 1.* $\tau^*_{PL} > \tau^*_{RL} \geq \tau^*_{PC} > \tau^*_{RC}$ *iff* $\beta \leq \hat{\beta}\left(\bar{\alpha}, \frac{y_P}{y_R}\right)$.

- *Case 2.* $\tau^*_{PL} > \tau^*_{PC} \geq \tau^*_{RL} > \tau^*_{RC}$ *iff* $\beta \geq \hat{\beta}\left(\bar{\alpha}, \frac{y_P}{y_R}\right)$.

   *where* $\hat{\beta}\left(\bar{\alpha}, \frac{y_P}{y_R}\right) = \frac{\bar{\alpha}}{1-\bar{\alpha}}\left(\frac{y_P}{y_R}\right)^2$.

Proof: Consider that there is a majority of liberal voters $n_{RL} + n_{PL} \geq 1/2$. Then both parties propose $\bar{\alpha}$ in the second stage of the political game. From (2.1) and (2.2) it is straightforward that $\tau^*_{PL}(\bar{\alpha}) > \tau^*_{PC}(\bar{\alpha})$ and $\tau^*_{RL}(\bar{\alpha}) > \tau^*_{RC}(\bar{\alpha})$. However, the ordering in the comparison between $\tau^*_{PC}(\bar{\alpha})$ and $\tau^*_{RL}(\bar{\alpha})$ is not so evident. In particular $\tau^*_{RL}(\bar{\alpha}) \geq \tau^*_{PC}(\bar{\alpha})$ iff:

$$\frac{\bar{\alpha}Y}{4y_R^2} \geq \frac{\beta(1-\bar{\alpha})Y}{4y_P^2} \Leftrightarrow$$

$$\beta \leq \hat{\beta} = \frac{\bar{\alpha}}{1-\bar{\alpha}}\left(\frac{y_P}{y_R}\right)^2.$$

Therefore, there are two possible orders regarding the voters' optimal tax rates:

Case 1. $\tau^*_{PL} > \tau^*_{RL} \geq \tau^*_{PC} > \tau^*_{RC}$ iff $\beta \leq \hat{\beta}\left(\bar{\alpha}, \frac{y_P}{y_R}\right)$.

Case 2. $\tau^*_{PL} > \tau^*_{PC} \geq \tau^*_{RL} > \tau^*_{RC}$ iff $\beta \geq \hat{\beta}\left(\bar{\alpha}, \frac{y_P}{y_R}\right)$.

Case 1 describes a society where either there is a low level of income inequality and liberal voters are much more likely than conservatives to support public spending, or both. In this

scenario, ideology—defined as the different preferences for different public goods—describes two well-defined blocks of voters: the block of liberals who want a larger welfare state, and the block of conservatives who want a smaller one. In contrast, Case 2 represents the scenario in which conservatives' and liberals' support for public spending is alike and income inequality is high enough. In this case, income rather than ideological views shapes differences in preferences for public spending. The poor prefer greater public spending and the rich prefer less.

Taking this into account, along with the two possible cases of voters preferences stated in Lemma 1, I present the equilibrium policies resulting from the two-stage voting process in the following Proposition.

**Proposition 1** *If there is a liberal majority, i.e.* $n_{RL} + n_{PL} \geq 1/2$ *the policy implemented in equilibrium is the optimal policy for the rich liberal voter* $(\tau_{RL}^*(\bar{\alpha}), \bar{\alpha})$ *if* $\beta \leq \hat{\beta}\left(\bar{\alpha}, \frac{y_P}{y_R}\right)$, *and it is the optimal policy for the poor conservative voter* $(\tau_{PC}^*(\bar{\alpha}), \bar{\alpha})$ *otherwise.*

<u>Proof:</u> Consider that there is a majority of liberal voters $n_{RL} + n_{PL} \geq 1/2$. Then both parties propose $\bar{\alpha}$ in the second stage of the political game. According to Lemma 1, I have two cases regarding voter preferences. Given that voters' preferences are single-peaked and parties are office seekers, both parties propose the optimal policy of the median voter according to the median voter theorem. In Case 1, i.e. if $\beta \leq \hat{\beta}\left(\bar{\alpha}, \frac{y_P}{y_R}\right)$, the median voter is the liberal rich voter, since $n_{RL} + n_{PL} \geq 1/2$; and in Case 2, i.e if $\beta \geq \hat{\beta}\left(\bar{\alpha}, \frac{y_P}{y_R}\right)$, the median voter is the conservative poor voter, since $n_{PL} + n_{PC} \geq 1/2$. Hence, the policy implemented in equilibrium is $(\tau_{RL}^*(\bar{\alpha}), \bar{\alpha})$ if $\beta \leq \hat{\beta}\left(\bar{\alpha}, \frac{y_P}{y_R}\right)$, and $(\tau_{PC}^*(\bar{\alpha}), \bar{\alpha})$ otherwise.

The proposition above underlines a change in the type of the decisive voter, depending on the level of the intensity of conservatives' support for public spending and the level of income inequality. In particular the decisive voter depends on whether the intensity of conservatives' support for public spending is greater than the threshold function $\hat{\beta}\left(\bar{\alpha}, \frac{y_P}{y_R}\right)$. Notice that this function is decreasing in income inequality (i.e. increasing in $\frac{y_P}{y_R}$) and increasing in the share of tax revenues devoted to the liberal policy ($\bar{\alpha}$).

Fig 1 pictures the decisive voter for any combination of income inequality and conservative preferences for public spending for the case of $\bar{\alpha} = 0.75$, i.e. $\alpha \in [0.25, 0.75]$.

According to Fig 1, the liberal rich becomes the decisive voter if there is either little support for public spending among conservatives (for some given income inequality) or low income inequality (again, for some fixed support for public spending). Therefore a winning candidate's campaign would target the liberal rich voter in either a context of low income inequality or a context of little support for public spending among conservative voters. This is because liberal voters (rich and poor) form a majority that prefers a higher tax rate and public spending more than conservative voters do. Since preferences over the tax rate are single-peaked, and following the median voter theorem, parties maximize the probability of winning by proposing the optimal policy for the liberal rich voter, who is the one with a more centered position in the majoritarian liberal coalition.

However, if conservative voters increase their support for public spending and income inequality becomes higher, then a winning candidate's campaign would target the conservative poor voter. This is because if $\beta \geq \hat{\beta}\left(\bar{\alpha}, \frac{y_P}{y_R}\right)$, the majoritarian coalition is now formed by (liberal and conservative) poor voters who want a higher public expenditure than rich voters. In order to maximize the probability of winning, both parties will offer the optimal policy of the more moderate voter in the coalition, who in this case is the conservative poor voter.

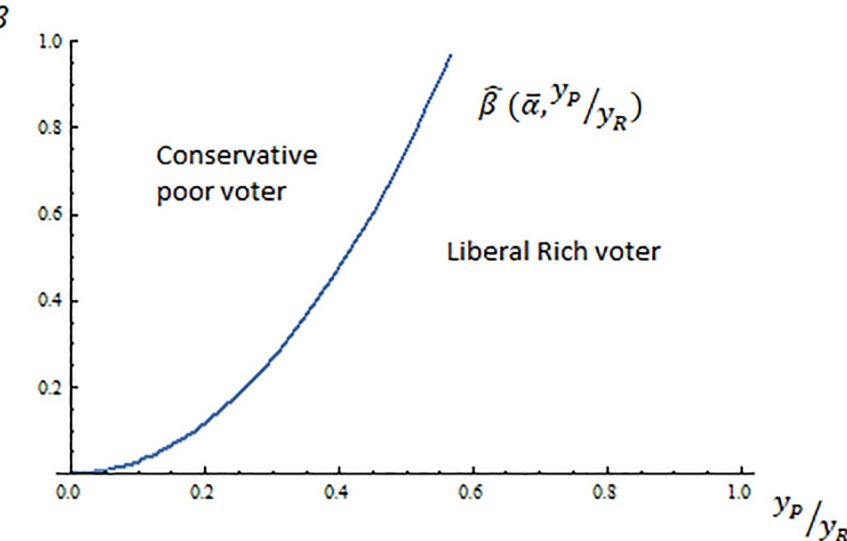

**Fig 1. The decisive voter.**

The results indicate that ideology divides the electorate and defines the winning policy proposal in elections when conservatives' support for public spending is much lower than liberal voters' support (and income inequality is not very high such that $\beta \leq \hat{\beta}\left(\bar{\alpha}, \frac{y_P}{y_R}\right)$). In this case, I expect ideology to define parties' strategies with a liberal party and a conservative party. Nevertheless the income level divides constituencies and is the key to winning the elections when the support for public spending is similar for conservatives and liberals and income inequality is high ($\beta \geq \hat{\beta}\left(\bar{\alpha}, \frac{y_P}{y_R}\right)$). In this case, parties are expected to be either pro-poor or pro-rich and the decisive voter is the conservative poor.

Another interesting exercise is to check how the decisive voter changes for different democracies depending on their tradition of supporting liberal policies. That is, how the decisive voter respond to changes in $\bar{\alpha}$. Since $\hat{\beta}\left(\bar{\alpha}, \frac{y_P}{y_R}\right)$ is increasing in the share of tax revenues devoted to the liberal policy ($\bar{\alpha}$), democracies with a bigger share of tax revenues devoted to liberal policies need to undergo a larger increase in income inequality to change the decisive voter from the rich liberal to the conservative poor. To illustrate this, Fig 2 presents the effect of an increase in $\bar{\alpha}$ (from $\bar{\alpha}$ to $\bar{\alpha}'$) on the regions defining where the decisive voter is either the liberal rich or the conservative poor. An increase $\bar{\alpha}$ reduces (enlarges) the region where the conservative (liberal) rich (poor) is the decisive voter. According to my theory, given two democracies with the same conservatives' support for public spending and the same level of income inequality, it is more likely to have a change in the decisive voter from the liberal rich to the conservative poor in democracies with less of a tradition of liberal policies.

Results for the scenario in which there is a majority of Conservative voters $n_{RC} + n_{PC} \geq 1/2$ are quite different regarding the type of decisive voter. In a nutshell, if the support of public spending is low enough the decisive voters is always the Conservative poor. However if the support for public spending is similar for conservatives and liberals, ideology divides constituencies and the Conservative rich voter becomes the decisive voter. The whole analysis and casuistry can be seen in the S1 Appendix.

Until now I only study the political equilibria regarding the type of the decisive voter. Next, I analyze the tax rate chosen in equilibrium and therefore the size of the welfare state.

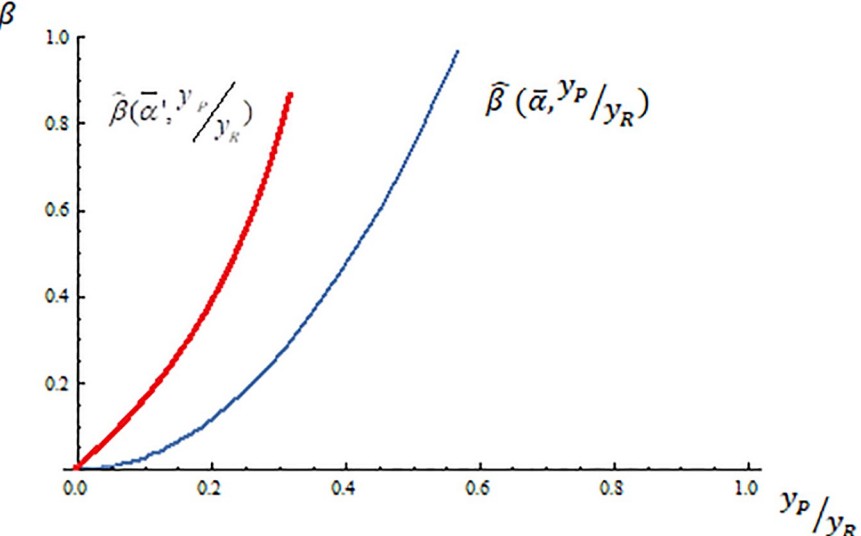

**Fig 2. The effect of $\bar{\alpha}$ in the decisive voter.**

## 2.2 The size of the welfare state

First, notice that since the government budget constraint is assumed to be balanced, the tax rate implemented determines the size of the welfare state. By Proposition 1 the size of the welfare state is given by the optimal tax rate for either the liberal rich or the conservative poor. From (2.1) and (2.2) the optimal tax rate for liberal rich is decreasing in income inequality and does not depend on conservatives' support of public spending ($\beta$). Differently the optimal tax rate for conservative poor is increasing in both income inequality and $\beta$. Moreover, by Proposition 1 we know that if beta is small enough ($\beta \leq \hat{\beta}\left(\bar{\alpha}, \frac{y_P}{y_R}\right)$) then the implemented tax rate is the optimal tax rate for the liberal rich (a constant function of $\beta$) and it is the optimal one for the conservative poor (an increasing function of $\beta$), otherwise. Fig 3 pictures the size of the welfare state depending on the Conservatives' support for public spending for the case of $\bar{\alpha} = 0.75$ and $\frac{y_P}{y_R} = 0, 8$.

Next, I deal with the remained question of how income inequality affects the the size of the welfare state.

**Proposition 2** *If there is a liberal majority, i.e. $n_{RL} + n_{PL} \geq 1/2$, an increase in income inequality reduces (increases) the size of the welfare state if the support for public spending among conservative voters $\beta$ is low (high) enough.*

<u>Proof</u>: From (2.1) and (2.2) and Proposition 1, the implemented tax rate in equilibrium is a constant function in $\beta$ equals to $\frac{\bar{\alpha}Y}{4y_R^2}$ if $\beta \leq \hat{\beta}\left(\bar{\alpha}, \frac{y_P}{y_R}\right)$ and it is a linear function in $\beta$ with a positive slope equal to $\frac{(1-\bar{\alpha})Y}{4y_P^2}$ if $\beta \geq \hat{\beta}\left(\bar{\alpha}, \frac{y_P}{y_R}\right)$. Consider an increase in income inequality such that $\frac{y_P}{y_{R1}} \leq \frac{y_P}{y_{R0}}$. First notice that since $\frac{\bar{\alpha}Y}{4y_R^2}$ is decreasing in income inequality (decreasing in $y_R$ and increasing in $y_P$) the implemented tax rate in equilibrium is now lower if $\beta \leq \hat{\beta}\left(\bar{\alpha}, \frac{y_P}{y_{R1}}\right)$. Otherwise if $\beta \geq \hat{\beta}\left(\bar{\alpha}, \frac{y_P}{y_{R1}}\right)$ the implemented tax rate in equilibrium is a linear function in $\beta$ with a positive slope equals to $\frac{(1-\bar{\alpha})Y}{4y_P^2}$. This slope is increasing with income inequality. Hence, there

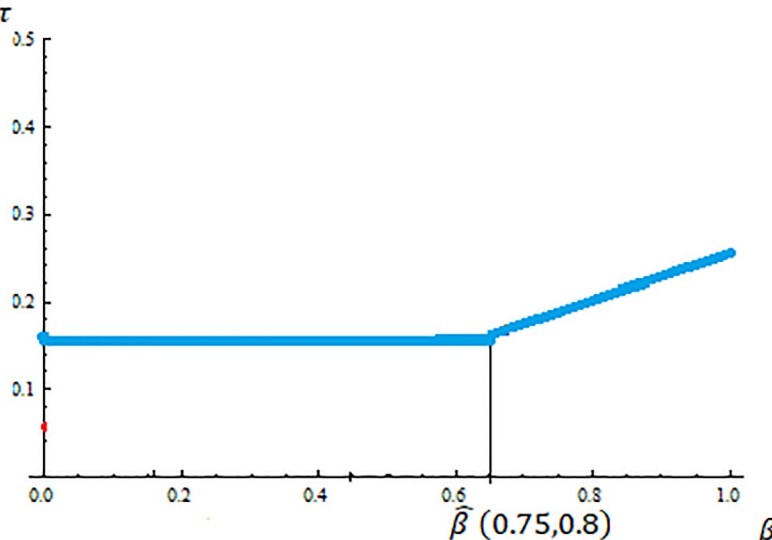

**Fig 3. The size of the welfare state.**

always exist a $\beta^*$ such that an increase in income inequality reduces (increases) the size of the welfare state if the support for public spending among conservative voters $\beta \leq \beta^*$ ($\beta \geq \beta^*$).

Proposition 2 underlines that, unlike the traditional models on redistributive politics [1, 11, 12], an increase in income inequality does not necessarily imply an increase in the tax rate chosen by majority voting. I obtain this result only if the support for public spending is similar for conservatives and liberals. However, if conservatives' support for public spending is low enough, an increase in income inequality reduces the implemented tax rate, and therefore the size of the welfare state. Fig 4 illustrate this result showing the implemented tax rates in equilibrium for the case of $\bar{\alpha} = 0.75$ and an increase of income inequality from $\frac{y_P}{y_R} = 0,8$ to $\frac{y_P}{y_R} = 0,4$.

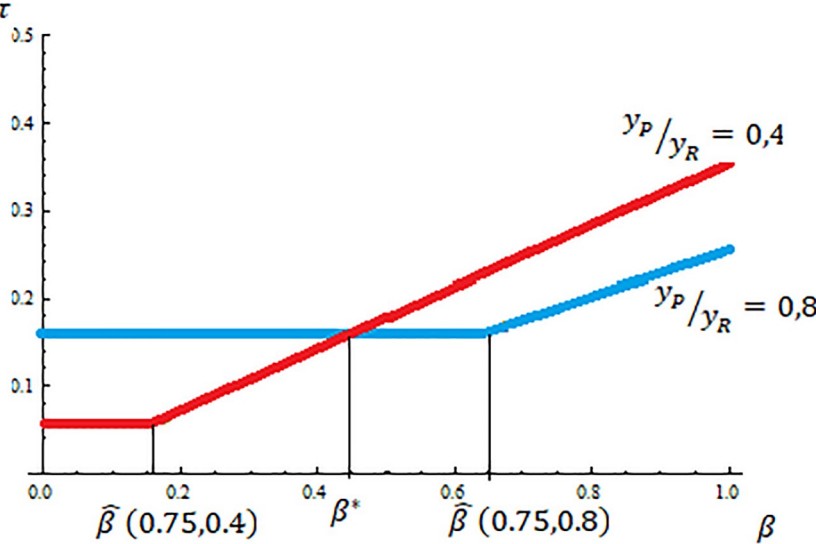

**Fig 4. The effect of income inequality in the size of the welfare state.**

How this effect depends on the size of the increase in income inequality? Notice that as the size of the increase in income inequality is larger both the optimal tax rate for the liberal rich $\tau_{RL}^*$ and $\hat{\beta}\left(\bar{\alpha}, \frac{y_P}{y_R}\right)$ become smaller. Also the slope of the optimal tax rate for the conservative poor $\tau_{PC}^*$ with respect to $\beta$ becomes higher. Therefore, using Fig 4, the higher the size of the increase in income inequality the lower the level of $\beta^*$ (that is the more likely to have an increase in the tax rate implemented).

The result underlined by Proposition 2 also have important implications to empirically study the relationship between income inequality and the size of the welfare state. In particular my model suggests that an interaction term of two variables such as the income inequality and the support of public spending among conservatives voters (relative to liberals) should be included in any regression to analyze the size of the welfare state.

## 3 Concluding remarks

Studies based on the median voter theorem, e.g. [1, 11, 12], state that an increase in income inequality leads the median voter to demand a higher tax rate and thus, a higher size of the welfare state. According to the empirical evidence from the 1970s to today, this theory is not well supported, particularly in the US (see, e.g., Roemer, 1998, and citations therein). This paper attempts to solve this puzzle by offering an alternative theory. Based on new empirical findings showing that preferences for taxation depend on the nature of the policies financed with tax revenues [7–10], I propose a simple Downsian two-party political competition framework in which voters differ in income (rich and poor) and in views about what kind of policy should be financed with tax revenues and how much government should spend on that policy (liberal and conservative).

I find that the reduction of the size of the welfare state depends crucially on the extent to which conservatives differ from liberals in preferences for public spending. In particular, the model predicts that large income inequality is compatible with low tax rates chosen by majority voting if the support for public spending is low among conservative poor voters. In this case, ideology rather than income divides the electorate, making the liberal rich voter the decisive voter.

Finally, the main result of the paper suggests that liberal rich elites would tend to coordinate strategically to campaign in favor of making poor conservatives more opposed to public spending in liberal democracies. This way they avoid elections in which parties' constituencies are divided by voters' income level, and face elections in which they become the decisive voters.

## Supporting information

**S1 Appendix. Conservative majority.**
(PDF)

## Author Contributions

**Conceptualization:** Ángel Solano-García.

**Formal analysis:** Ángel Solano-García.

**Resources:** Ángel Solano-García.

**Writing – original draft:** Ángel Solano-García.

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
