## [Decision Letter · Decision Letter 0]

29 Jun 2022

PONE-D-22-13644Income inequality and voters’ support for government intervention. A simple political modelPLOS ONE

Dear Dr. Solano Garcia,

Thank you for submitting your paper to PLOS ONE. I sent your paper to three referees, of which just two have now returned their reports and recommendations. To insure a timely response, I have decided to wait no further and proceed with my decision. You can find the reports in the Editorial Manager PLOS ONE platform. Both referees like the general idea but, at the same time, express major concerns of some features of your theoretical exercise. As a result, one requests a major revision, while the other recommends rejection. I also had my own reading of your paper and have finally decided to follow the former recommendation, with the caveat that I expect a thorough revision compared with the original version for your submission to have chance of success. 

Let me here briefly discuss here the reasons for my decision. Both referees object (and I fully agree with both of them) that your model is far too mechanical to have some real explanatory power in that the key variables that you employ (with specific reference with income inequality) are fixed exogenously beforehand. This is the main criticism that your revision needs to address in the first place for your contribution to have some convincing predicting power.

To summarize: I look forward for an extensively revised version of the paper that properly takes advantage of the very competent suggestions included in the two reports. Both reports contain substantive critiques that need to be addressed properly as a necessary condition for consideration for publication.  

We look forward to receiving your revised manuscript.

Kind regards,

giovanni ponti

Academic Editor

PLOS ONE

Journal Requirements:

Reviewers' comments:

Reviewer's Responses to Questions

**Comments to the Author**

1. Is the manuscript technically sound, and do the data support the conclusions?

Reviewer #1: Yes

Reviewer #2: Yes

2. Has the statistical analysis been performed appropriately and rigorously? 

Reviewer #1: N/A

Reviewer #2: N/A

3. Have the authors made all data underlying the findings in their manuscript fully available?

Reviewer #1: Yes

Reviewer #2: Yes

4. Is the manuscript presented in an intelligible fashion and written in standard English?

Reviewer #1: Yes

Reviewer #2: Yes

5. Review Comments to the Author

Reviewer #1: Referee report for PLOS-ONE:

“Income inequality and voters’ support for government intervention: a simple political model

(PONE-D-22-13644)

Summary

The paper proposes a theoretical model to study the role of income inequality and support for public spending among conservative voters on the tax rate chosen by majority voting. In this model individuals differ in income (poor and rich) and preferences for public services (liberal and conservative). Government choices are a proportional income tax rate and the share of tax financing the liberal public good (which, in turn, determines the size of the conservative public good). Voters vote first for income tax rate and then for the share of tax revenue allocated to finance each public good. The author solves backwards, using the subgame perfect Nash equilibrium. The main result is that, under a liberal majority in the population, the liberal rich is the decisive voter if there is low income inequality and/or the support for public spending among conservatives is much lower than among liberals. If there high income inequality and/or the support for public spending among conservatives is similar to that than among liberals, then the poor conservative is the decisive voter.

Evaluation

In my opinion this paper addresses an important question. However, I have some reservations about the implementation of the idea.

Main comments:

The author claims in the introduction and conclusions that income inequality depends on support for public spending among conservatives (and income inequality? See for instance, third paragraph in page 3). This is confusing: in this paper income inequality is an exogenous variable which affects, together with the support for public spending among conservatives, the type of decisive voter.

The main result should be clarified and better stated. As it stands now it is the combination of two variables (income inequality and preferences for public spending among conservatives) that determines the decisive voter (either liberal if the former is low and the latter is also low, or conservative in case of high income inequality and/or high support for public spending among conservatives). Whether or not the optimal tax rate among liberals is larger than the one among conservatives depends on the combination of the abovementioned variables. Therefore, it is not clear how optimal taxes can be compared in absolute terms which seems to be the conclusion derived from Proposition 1 as mentioned in the Abstract, Introduction and Concluding remarks.

Other comments

Abstract and Introduction

I suggest clarifying what “ideology” means in the abstract while first introducing the term. Also related to this point, I would stick to the “support for public spending” term while referring to ideology and differences between liberal and conservatives: “support for government intervention” might be confusing as government intervention can also be measured through tax size (which is also another endogenous variable in the model)

In my view, the author should add some references to better justify claims related to the strategies followed by rightist parties to attract poor voters (first paragraph in the Introduction).

I suggest moving to the model justifications and empirical support for ideology (why liberals vs conservatives) and the lower support for public spending among conservatives.

The Model

The selection of the scenario where there is a liberal majority deserves additional discussion. I suggest better justifying it, by mentioning empirical evidence. In addition, the intuition of results in the other scenario should be commented in the main text to discuss the extent to which the main result in the paper depends on this assumption.

A definition of a threshold β ^((α,) ®y_P/y_R)=α ®/(1-α ® ) 〖(y_P/y_R )〗^2 would facilitate readership. The discussion of proposition 1 should then be re-written, together with comments on comparative statics around this threshold.

Could figures 1 and 2 be done by using the same definition of income inequality used in the rest of the paper?

Minor comments:

I suggest being consistent with the use of “&” or “and” in the citing references.

Figure 1 is plotted for “some” alpha between 0.25 and 0.75 (i.e., the parameter is fixed).

There are several typos, for instance: “that” instead of “than” in line 3 page 4; some “.” and “,” should be deleted in proof of Proposition 1.

Reviewer #2: I find that this paper investigates an extremely important issue for our democratic countries. I think that, however, it does not touch the main issue: why, in the last decades, the system evolved such that the two dimensions (redistribution vs ideology) now interlink more? This is the key question we are facing now.

The model provide an extremely simplified framework in which all the main quantities are exogenously given, and thus no real explanation of the phenomena can be provided. I personally see this peace of research as an interesting starting point for research on this topic but I hardly see it as a full paper.

Considering condition below eq (2.2), if I am not wrong, this should be tau_c(alpha_under)=(beta (1-alpha_under)/alpha^bar) tau_L(alpha^bar). What also happens if beta=1? I also think that the author should also improve the understanding of the conditions of the two cases.

6. PLOS authors have the option to publish the peer review history of their article (what does this mean?). If published, this will include your full peer review and any attached files.

Reviewer #1: No

Reviewer #2: No

---

## [Author Response · Author response to Decision Letter 0]

9 Aug 2022

Dear Prof. Ponti,

We thank you for the opportunity to revise my paper. I have indeed followed all the suggestions made and we think the paper has clearly improved. In particular, I clarify that the objective of the paper is to better understand how pre-tax income inequality affect the size of the welfare state. To do so, in addition to follow all reviewers’ suggestions, I have changed the title of the paper and include a new subsection. 

The main result of the model has an important implication to the future empirical strategies to estimate the effect of income inequality in the size of the welfare state. According to this study, any regression to estimate the size of the welfare state should include, among the usual explanatory variables, an interaction term of two variables such as the pre-tax income inequality and the support of public spending among conservatives voters (relative to liberals). 

I have enclosed a copy of the revised manuscript along with the response to both reviewers.

We hope that these changes will make the goal of the paper more clear. Otherwise, I am open to change the structure of the paper if you consider it is essential for the sake of clarity.

Sincerely,

Ángel Solano

Reply to Reviewer #1

Dear Reviewer,

Thank you for your thoughtful comments on our manuscript. We believe that now the main goal of the paper is clearer. We have rewritten the title, abstract, introduction and conclusions and included a new subsection to account for your concerns. 

Let us now detail how we have addressed your specific comments, keeping the same order as in your report:

1. The author claims in the introduction and conclusions that income inequality depends on support for public spending among conservatives (and income inequality? See for instance, third paragraph in page 3). This is confusing: in this paper income inequality is an exogenous variable which affects, together with the support for public spending among conservatives, the type of decisive voter.

The main result should be clarified and better stated. As it stands now it is the combination of two variables (income inequality and preferences for public spending among conservatives) that determines the decisive voter (either liberal if the former is low and the latter is also low, or conservative in case of high income inequality and/or high support for public spending among conservatives). Whether or not the optimal tax rate among liberals is larger than the one among conservatives depends on the combination of the abovementioned variables. Therefore, it is not clear how optimal taxes can be compared in absolute terms which seems to be the conclusion derived from Proposition 1 as mentioned in the Abstract, Introduction and Concluding remarks.

Thank you for your comment. I agree that it is confusing to use the term income inequality as an exogenous variable and an endogenous variable at the same time. I did that because I was referring to the pre-tax income inequality to the exogenous variable (yP/yR) and to the post-tax income inequality to the endogenous variable. Implicitly, I assumed that the size of the public good devoted to liberal purposes reduces post-tax income inequality (although it is not modeled as an in-cash social transfer to the poor). Of course this is very confusing and I want to thank you to point this out. 

Following your advice I stick to the pre-tax income inequality definition (as an exogenous variable) through the whole paper and analyze its effect (joint with effect of preferences for public spending) on the decisive voter. Also I clarify the main point of the paper as you suggest. 

However, I also focus on the effect of the exogenous variables in size of the welfare state (given by the tax rate chosen and the share of tax revenue devoted to provide the liberal public good). This is because this paper tries to contribute to give an answer to the puzzle between the empirical findings and the traditional theories regarding the relationship between pre-tax income inequality and the size of the welfare state. To do so I include a new subsection “The size of the welfare state” with a new Proposition, and a new figure to show the relationship between pre-tax income inequality, voters’ support for public spending and the size of the welfare state. I hope that all this changes helps to make the main points of the paper clearer. 

2. I suggest clarifying what “ideology” means in the abstract while first introducing the term. 

Done. Thank you for your comment.

3. Also related to this point, I would stick to the “support for public spending” term while referring to ideology and differences between liberal and conservatives: “support for government intervention” might be confusing as government intervention can also be measured through tax size (which is also another endogenous variable in the model)

Thank for suggesting this. I agree with you and I have done this through all the text.

4. In my view, the author should add some references to better justify claims related to the strategies followed by rightist parties to attract poor voters (first paragraph in the Introduction).

Done. Thank you for your comment.

5. I suggest moving to the model justifications and empirical support for ideology (why liberals vs conservatives) and the lower support for public spending among conservatives.

Done. Thank for suggesting this.

6. The selection of the scenario where there is a liberal majority deserves additional discussion. I suggest better justifying it, by mentioning empirical evidence.

Done. Thank you for your comment.

7. In addition, the intuition of results in the other scenario should be commented in the main text to discuss the extent to which the main result in the paper depends on this assumption.

Done. Thank for suggesting this.

8. A definition of a threshold β ^((α,) ®y_P/y_R)=α ®/(1-α ® ) 〖(y_P/y_R )〗^2 would facilitate readership. The discussion of proposition 1 should then be re-written, together with comments on comparative statics around this threshold.

Done. Thank you for your comment.

9. Could figures 1 and 2 be done by using the same definition of income inequality used in the rest of the paper?

Yes I did this with Figure 1. However, and thanks to the definition of the new threshold function beta hat, we do need to use Figure 2 to explain how affect the share of tax revenues devoted to liberal policies to the decisive voter. Following the referee’s advice we provide this explanation using the threshold proposed. 

Minor comments:

I suggest being consistent with the use of “&” or “and” in the citing references.

Done

Figure 1 is plotted for “some” alpha between 0.25 and 0.75 (i.e., the parameter is fixed).

Thank you for this comment. Ie add that Figure 1 pictures the decisive voter for the case of α upper bar =0.75 which implies that α is between (0,25, 0,75) (given that α lower bar plus α upper bar equals to one)

There are several typos, for instance: “that” instead of “than” in line 3 page 4; some “.” and “,” should be deleted in proof of Proposition 1.

Done

Reply to Reviewer #2

Dear Reviewer,

Thank you for your thoughtful comments on our manuscript. We believe than now our empirical results are more robust. We have included two new sections (although in an Appendix to not increase too much the length of the manuscript) to account for your concerns.

Let us now detail how we have addressed your specific comments, keeping the same order as in your report:

1. I find that this paper investigates an extremely important issue for our democratic countries. I think that, however, it does not touch the main issue: why, in the last decades, the system evolved such that the two dimensions (redistribution vs ideology) now interlink more? This is the key question we are facing now.

I agree with the referee that the question: why, in the last decades, redistribution and ideology interlink more? is very interesting. I also agree that this paper is not addressing this point. However we refer in the text some interesting studies that they do such as Gethin et al. 2021. We use this and other studies to justify our assumptions. 

Our paper focus in an old puzzle: according to traditional literature in redistributive politics there is a positive relationship between the size of the welfare state and the level of income inequality. However empirical findings do not confirm this result. We propose and alternative explanation. Moreover, the findings suggest that an interaction term of two variables such as the pre-tax income inequality and the support of public spending among conservative voters (relative to liberal) should be included in any regression to analyze the size of the welfare state. 

2 The model provide an extremely simplified framework in which all the main quantities are exogenously given, and thus no real explanation of the phenomena can be provided. I personally see this peace of research as an interesting starting point for research on this topic but I hardly see it as a full paper.

I agree with referee that the model is very striped down. However we clarify that there are two endogenous variables which are the tax rate and the share of tax revenues devoted to the liberal public good. Multidimensionality in political competition models makes quite complicate the calculus of equilibria (see Roemer 2001). A solution for this was used in Alesina et al (1999) adopting a two-stage voting process to avoid the problem of multidimensionality when aggregating preferences. I use this method for the sake of clarity.

3. Considering condition below eq (2.2), if I am not wrong, this should be tau_c(alpha_under)=(beta (1-alpha_under)/alpha^bar) tau_L(alpha^bar). What also happens if beta=1? 

Thank you for this comment. Since α(upper bar) + α(lower bar)=1, we obtain the condition (2.3). Moreover a paragraph is added explaining the results for the case of beta = 1 (see forth paragraph in page 5). 

4. I also think that the author should also improve the understanding of the conditions of the two cases.

Thank for suggesting this. Using a definition of the new threshold function beta hat suggested by the other referee we have rewritten this part. I believe it now less demanding for the reader. 

Reference

ROEMER, J. E. (2001). Political Competition: Theory and Applications. Harvard University Press.

---

## [Decision Letter · Decision Letter 1]

19 Sep 2022

PONE-D-22-13644R1Income inequality, voters’ support for public spending and the size of the welfare state. A simple political modelPLOS ONE

Dear Dr. Solano Garcia,

Thank you for submitting your manuscript to PLOS ONE. I have sent your revised paper to the same set of reviewers as your original submission, one of which has now responded. S/he seems quite satisfied by your revision and recommends to accept the paper subject to fixing some additional (minor) comments. I am happy to follow this recommendation. Please submit your revised manuscript by Nov 03 2022 11:59PM. If you will need more time than this to complete your revisions, please reply to this message or contact the journal office at plosone@plos.org. Please include the following items when submitting your revised manuscript:A rebuttal letter that responds to each point raised by the academic editor and reviewer(s). You should upload this letter as a separate file labeled 'Response to Reviewers'.A marked-up copy of your manuscript that highlights changes made to the original version. You should upload this as a separate file labeled 'Revised Manuscript with Track Changes'.An unmarked version of your revised paper without tracked changes. You should upload this as a separate file labeled 'Manuscript'.If applicable, we recommend that you deposit your laboratory protocols in protocols.io to enhance the reproducibility of your results. Protocols.io assigns your protocol its own identifier (DOI) so that it can be cited independently in the future. For instructions see: https://journals.plos.org/plosone/s/submission-guidelines#loc-laboratory-protocols. Additionally, PLOS ONE offers an option for publishing peer-reviewed Lab Protocol articles, which describe protocols hosted on protocols.io. Read more information on sharing protocols at https://plos.org/protocols?utm_medium=editorial-email&utm_source=authorletters&utm_campaign=protocols.

We look forward to receiving your revised manuscript.

Kind regards,

Giovanni Ponti

Academic Editor

PLOS ONE

Journal Requirements:

Reviewers' comments:

Reviewer's Responses to Questions

**Comments to the Author**

1. If the authors have adequately addressed your comments raised in a previous round of review and you feel that this manuscript is now acceptable for publication, you may indicate that here to bypass the “Comments to the Author” section, enter your conflict of interest statement in the “Confidential to Editor” section, and submit your "Accept" recommendation.

Reviewer #1: All comments have been addressed

2. Is the manuscript technically sound, and do the data support the conclusions?

Reviewer #1: Yes

3. Has the statistical analysis been performed appropriately and rigorously? 

Reviewer #1: N/A

4. Have the authors made all data underlying the findings in their manuscript fully available?

Reviewer #1: Yes

5. Is the manuscript presented in an intelligible fashion and written in standard English?

Reviewer #1: Yes

6. Review Comments to the Author

Reviewer #1: Referee report for PLOS-ONE:

“Income inequality, voters’ support for public spending and the size of the welfare state. A simple political model”

(PONE-D-22-13644R1)

Evaluation

The author made substantial improvements to the paper after the submission. The main result is clearer and better stated now: the role of income inequality (as an exogenous variable) in the main finding has now been clarified. I just think the author should still make some additional revisions before publishing the paper.

1. To the extent that income inequality is no longer considered as an outcome variable, I think there is no need to name it as “pre-tax” income inequality (there is no such a thing as post-tax income inequality in the current version). Therefore, I suggest revising the text and systematically use “income inequality” (as it stands it might be confusing the use of pre-tax income inequality in some parts while just income inequality in others).

2. I suggest defining income inequality (using model notation) the first time it is mentioned in the text. Even though it is quite simple, in my view it would help readership considering it is one of the two key variables in the analysis.

3. The intuition of Proposition 1 (last paragraph in page 7) should be revised: the liberal rich becomes the decisive voter if there is either little support for public spending among conservatives (for some given income inequality) or low income inequality (again, for some fixed support for public spending).

4. As it stands, results mentioned in the third paragraph in page 8 could be interpreted as derived from a second order derivative in beta hat (“According to my theory, the type of decisive voter is less reactive to changes in income inequality in consolidated liberal democracies.”). I suggest clarifying this point.

5. The impact of income inequality on the liberal rich and conservative poor optimal tax rates should be shown in some more detail considering its relevance in the analysis.

6. Is it possible to provide and explicit function for beta*? It would be interesting to provide additional details on this threshold.

7. Finally, the paper could really benefit from thorough editing. There are complicated sentences and constructions, as well as grammar issues, that make the paper difficult to follow. Just to give one example, the alternate use of “I” or “we”; p. 6, line 1 (The International Social Survey Programme also obtainS…). is another example of this. Other small errors appear throughout (p. 7 "the the intensity of conservatives’…"; p. 9 “(..) foR the case…" etc.).

7. PLOS authors have the option to publish the peer review history of their article (what does this mean?). If published, this will include your full peer review and any attached files.

Reviewer #1: No

---

## [Author Response · Author response to Decision Letter 1]

28 Sep 2022

Dear Reviewer,

We would like to thank you for your very useful comments that have improved the paper considerably. We now detail how we address the minor changes you propose.

1. To the extent that income inequality is no longer considered as an outcome variable, I think there is no need to name it as “pre-tax” income inequality (there is no such a thing as post-tax income inequality in the current version). Therefore, I suggest revising the text and systematically use “income inequality” (as it stands it might be confusing the use of pre-tax income inequality in some parts while just income inequality in others).

Done. Thank you for your suggestion.

2. I suggest defining income inequality (using model notation) the first time it is mentioned in the text. Even though it is quite simple, in my view it would help readership considering it is one of the two key variables in the analysis.

Done. Thank you for suggesting this.

3. The intuition of Proposition 1 (last paragraph in page 7) should be revised: the liberal rich becomes the decisive voter if there is either little support for public spending among conservatives (for some given income inequality) or low income inequality (again, for some fixed support for public spending).

Thank you for your suggestion. I clarified this point using Figure 1.

4. As it stands, results mentioned in the third paragraph in page 8 could be interpreted as derived from a second order derivative in beta hat (“According to my theory, the type of decisive voter is less reactive to changes in income inequality in consolidated liberal democracies.”). I suggest clarifying this point.

Thank you for suggesting this. I clarified that it is an exercise to see how the decisive voter respond to changes in . To do this I introduce the Figure 2.

5. The impact of income inequality on the liberal rich and conservative poor optimal tax rates should be shown in some more detail considering its relevance in the analysis.

Thank you for your suggestion. I made explicit the effect of income inequality on the liberal rich and conservative poor optimal tax rates. I also extended the whole subsection and introduced Figure 3 to explain in a clearer way the impact of income inequality on the size of the welfare state. 

6. Is it possible to provide and explicit function for beta*? It would be interesting to provide additional details on this threshold.

Thank you for suggesting this. The explicit function for depends on the size of the variation of income inequality ( a reduction of ). An increase of income inequality can be done through a variation of income levels and such that . This increase could rise or lower the total income level from Y to Y’. Unfortunately, it is not possible to find an explicit expression of depending on the level of income inequality without normalizing either or , and therefore without assuming and effect on Y. However, it is possible to analyze the effect of the size of an increase in income inequality on without using its explicit function. As I mention in the proof of Proposition 2 as the size of the increase in the income inequality is larger both the optimal tax rate for the liberal rich and become smaller. Also the slope of the optimal tax rate for the conservative poor with respect to becomes higher. Therefore, using Figure 4, the higher the size of an increase in income inequality the lower the level of (that is the more likely to have an increase in the tax rate implemented). I introduce this result in the text.

7. Finally, the paper could really benefit from thorough editing. There are complicated sentences and constructions, as well as grammar issues, that make the paper difficult to follow. Just to give one example, the alternate use of “I” or “we”; p. 6, line 1 (The International Social Survey Programme also obtainS…). is another example of this. Other small errors appear throughout (p. 7 "the the intensity of conservatives’…"; p. 9 “(..) foR the case…" etc.).

Done. Thank you for suggesting this.

---

## [Editor Report · Decision Letter 2]

24 Oct 2022

Income inequality, voters’ support for public spending and the size of the welfare state. A simple political model

PONE-D-22-13644R2

Dear Dr. Solano Garcia,

We’re pleased to inform you that your manuscript has been judged scientifically suitable for publication and will be formally accepted for publication once it meets all outstanding technical requirements.

Kind regards,

Giovanni Ponti

Academic Editor

PLOS ONE